# Prediction of Intrinsically Disordered Proteins Using Machine Learning Based on Low Complexity Methods

Xingming Zeng [1], Haiyuan Liu [1,*] and Hao He [2]

1 Tianjin Key Laboratory of Optoelectronic Sensor and Sensing Network Technology, School of Electronic Information and Optical Engineering, Nankai University, Tianjin 300350, China; zxm18235272177@163.com
2 Department of Communication Engineering, School of Electronic Information, Hebei University of Technology, Tianjin 300400, China; 2020038@hebut.edu.cn
* Correspondence: liuhaiyuan@nankai.edu.cn

**Abstract:** Prediction of intrinsic disordered proteins is a hot area in the field of bio-information. Due to the high cost of evaluating the disordered regions of protein sequences using experimental methods, we used a low-complexity prediction scheme. Sequence complexity is used in this scheme to calculate five features for each residue of the protein sequence, including the Shannon entropy, the Topo-logical entropy, the Permutation entropy and the weighted average values of two propensities. Particularly, this is the first time that permutation entropy has been applied to the field of protein sequencing. In addition, in the data preprocessing stage, an appropriately sized sliding window and a comprehensive oversampling scheme can be used to improve the prediction performance of our scheme, and two ensemble learning algorithms are also used to verify the prediction results before and after. The results show that adding permutation entropy improves the performance of the prediction algorithm, in which the MCC value can be improved from the original 0.465 to 0.526 in our scheme, proving its universality. Finally, we compare the simulation results of our scheme with those of some existing schemes to demonstrate its effectiveness.

**Keywords:** intrinsically disordered proteins; machine learning; permutation entropy; computational complexity

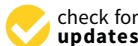



## 1. Introduction

As the highest content of organic compounds in human body, protein is the main bearer of human life activities, The "Amino Acid Sequence—3-Dimensional Structure—Protein Function" paradigm of protein was generally accepted [1]. However, in the past few decades, it has been found that not all proteins have a fixed three-dimensional structure in the whole sequence, and a protein lacking a specific three-dimensional structure has been continuously discovered by researchers [2]. These proteins lack a stable three-dimensional structure in at least one region and can also perform normal biological functions, so they are called intrinsically disordered proteins (IDPS). IDPS play an important role in physiological processes such as DNA transcription and translation [3]. Studies have shown that disordered proteins are associated with some major human diseases. For example, the lack of IDPS functionality may induce heart disease, Parkinson's disease, nerve tissue disease, cancer, etc. [4–8]. For example, the first pathogenic mutation in the SNCA gene, encoding for $\alpha$-synuclein was discovered in cases of familial Parkinson's disease [9], some of these point mutations cause Parkinson's with high penetrance. In addition, the severity of cognitive impairment in Alzheimer's disease was later shown to better correlate with low-molecular weight and soluble amyloid-beta aggregates, when amyloid-beta is highly disorganized in shape, it's actually less likely to stick together and form toxic clusters that lead to brain cell death [10]. A considerable number of biophysical studies have shown, type-2 diabetic islets are characterized by islet amyloid protein derived from islet amyloid peptide (IAPP), a protein co-expressed by beta cells with insulin that, when misfolded

and present in aggregated form, may lead to beta cell failure [11]. Therefore, more and more attention has been paid to the study of disordered proteins in recent years, and the research on the characteristics, functions and prediction of disordered proteins has also been greatly developed.

In the past few decades, there are various schemes for predicting IDPS that continue to emerge, and these methods are roughly divided into two categories: physicochemical-based and calculation-based. The first method is to detect IDPs by using amino acid propensity scale and physicochemical properties of protein sequence, such as GlobPlot [12], IUPred [13], FoldIndex [14] and IsUnstruct [15]. Compared with the physicochemical-based method, The second method distinguishes ordered and disordered proteins with positive samples and negative samples, effectively combines various features, and uses machine learning to make predictions, such as support vector machines (SVM), Naive Bayes (NB), K nearest neighbors (KNN) and decision trees (DT). These schemes include DISOPRED3 [16], SPINE-D [17], ESpritz [18] and MetaDisorder [19]. DISOPRED3 calculates the Position-Specific Substitution Matrix (PSSM) of all residues using three iterations of PSI-BLAST, and predicts the disordered regions and protein binding sites by using support vector machines as classifiers. SPINE-D uses a neural network to predict disorder regions, the algorithm makes a ternary prediction of all residues (ordered residues, short disordered region residues and long disordered region residues), then simplifies it to a binary prediction and trains both short disordered regions and long disordered regions. It is worth mentioning that the hybrid scheme based on various predictors can integrate a plurality of single schemes, and can better utilize the prediction advantages of different aspects of each single scheme, so as to improve the prediction accuracy. For example, MetaDisorder integrates predictors including DISOPRED2 [20], Globplot, IUpred, PrDOS [21], POO-DLE [22], DISPI [23], RONN [24], etc., So that its prediction results score higher than a single prediction result.

Most of the above IDPS prediction schemes use a large number of features, resulting in too high computational complexity to meet the requirements of making efficient predictions on a large number of data sets. Disordered proteins often have repetitive regions in their amino acid sequences, so they have lower sequence complexity than ordered proteins [25], We propose a new feature extraction scheme based on sequence complexity, which uses five features including Shannon entropy, topological entropy, permutation entropy and two amino acid preferences. Through the proposed preprocessing strategy, the selected features can better reflect the features of disordered regions, and has lower computational complexity than the existing prediction scheme. Finally, two boosting algorithm are used to verify the feasibility of the scheme.

The specific steps of the scheme are as follows:

Step 1: Download the latest 2209 intrinsically disordered protein sequences from DisProt (https://www.disprot.org/, accessed on 6 October 2021). The data set includes 1,217,223 amino acid residues, of which 995,189 residues are ordered and 222,034 residues are disordered.

Step 2: Since the nucleotides of the disordered protein coding gene are different from the ordered protein, the amino acid sequence of the disordered protein shows a more obvious bias. Compared with ordered proteins, disordered proteins have a lower content of hydrophobic residues. We corresponded the 20 amino acids to the numbers 0 and 1, which were used to calculate the permutation entropy.

Step 3: select a suitable sliding window, calculate the Shannon entropy, topological entropy, permutation entropy and two amino acid preferences of each residue, and finally acquire a $1,217,223 \times 5$ data set DIS2209.

Step 4: Due to the imbalance of the data samples, we performed three oversampling schemes on the data, and selected the comprehensive sampling with better performance. In addition, we used ten-fold cross-validation, using 90% of the DIS2209 data set randomly as the training set and 10% as the test set, and then using the grid search method to find the optimal parameter combination of the trainer, and finally calculating our Four

indicators needed: Sensitivity (Sens), Specificity (Spec), F1 score (F1), Matthews Correlation Coefficient (MCC).

Step 5: compare our schemes with the existing schemes.

## 2. Feature Selection and Preprocessing Process

The amino acid sequence of disordered proteins often has repeated regions, which is lower in sequence complexity than that of ordered proteins. According to this characteristic, we use Shannon entropy, topological entropy, permutation entropy and two amino acid preferences to describe the sequence complexity of proteins. A detailed description of these features follows.

### 2.1. Shannon Entropy

The Shannon entropy [26] is a standard measure for the order state of sequences and has been applied previously to protein sequences, it quantifies the probability density function of the distribution of values. If the length of a protein sequence $w$ is $N$, its Shannon entropy can be expressed as:

$$H_w = -\sum_{j=1}^{20} f_j \log_2 f_j \tag{1}$$

where $f_j (1 \leq j \leq 20)$ represents the frequency of the 20 amino acids in the sequence, and the formula can be expressed as:

$$f_j = \frac{\sum_{m=1}^{N} k(m)}{N} \tag{2}$$

When $m = j$, $k(m) = 1$, otherwise $k(m) = 0$.

### 2.2. Topological Entropy

Topological entropy [27–30] can also reflect the complexity of protein sequences very well, calculate the complexity function of the protein sequence $w$ of length $N$:

$$p_w(n) = |\{u : |u| = n\}| \tag{3}$$

$p_w(n)$ representing the total number of different $n$-length subwords of $w$, where $u$ is the subsequence of length $n$ $(1 \leq n \leq N)$ in the sequence. For example, given the sequence $w = \mathrm{AAATAA}$, $n = 2$, then the subsequence $w(u)$ of $w$ are $\{\mathrm{AA}, \mathrm{AT}, \mathrm{TA}\}$, so $p_w(n) = 3$.

For a sequence $w$ of length $N$ and a subsequence length of $n$, the following formula needs to be satisfied:

$$20^n + n - 1 \leq |w| < 20^{n+1} + (n+1) - 1 \tag{4}$$

Denote the segment consisting of $20^n + n - 1$ consecutive characters in the first paragraph of $w$ as $w_1^{20^n+n-1}$, namely:

$$w_1^{20^n+n-1} = w(1)w(2)\cdots w(20^n + n - 1) \tag{5}$$

Then the topological entropy of the protein sequence $w$ can be expressed as:

$$H_{top}(w) = \frac{\log_{20} p_{w_1^{20^n+n-1}}(n)}{n} \tag{6}$$

where $p_{w_1^{20^n+n-1}}(n)$ can be calculated by Equation (3), which represents the number of different subsequences of length n contained in the first segment of length $20^n + n - 1$ in sequence $w$. If fragment $w_1^{20^n+n-1}$ contains all subsequences of length $n$, then $H_{top}(w) = 1$, If fragment $w_1^{20^n+n-1}$ only contains one subsequence of length $n$, then $H_{top}(w) = 0$.

In order to further optimize the calculation result of topological entropy, we use the method of sequence traversal to calculate the topological entropy of each segment, and take the average value as the final topological entropy calculation result:

$$H_{top}(w) = \frac{1}{N - (20^n + n - 1) + 1} \sum_{l=1}^{N-(20^n+n-1)+1} \frac{\log_{20} p_{w_l^{20^n+n-1+l-1}}(n)}{n} \tag{7}$$

However, Equation (7) only needs to make the length of the protein sequence greater than 400 for the case of $n = 2$, which exceeds the length of most protein sequences. Therefore, according to the nature of the disordered protein that there are few hydrophobic residues [31], we map the sequence to 0 and 1: map hydrophobic (I,L,V,F,W,Y) residues to 1, and map other residues to 0, as shown in Table 1. Then the Equation (7) can be changed to:

$$H_{top}(w) = \frac{1}{N - (2^n + n - 1) + 1} \sum_{l=1}^{N-(2^n+n-1)+1} \frac{\log_2 p_{w_l^{2^n+n-1+l-1}}(n)}{n} \tag{8}$$

**Table 1.** Mapping values of topological entropy.

|                | **A** | **R** | **N** | **D** | **C** | **Q** | **E** | **G** | **H** | **K** |
|----------------|-------|-------|-------|-------|-------|-------|-------|-------|-------|-------|
| Mapping values | 0     | 0     | 0     | 0     | 0     | 0     | 0     | 0     | 0     | 0     |
|                | **M** | **P** | **S** | **T** | **I** | **L** | **F** | **W** | **Y** | **V** |
| Mapping values | 0     | 0     | 0     | 0     | 1     | 1     | 1     | 1     | 1     | 1     |

### 2.3. Permutation Entropy

In order to better highlight the complexity of protein sequences, we introduced the permutation entropy for the first time. Permutation entropy introduces the idea of permutation when calculating the complexity of reconstructed subsequences, it can be calculated for arbitrary real-world time series. Since the method is extremely fast and robust, it is preferable when there are huge data sets and no time for preprocessing and fine-tuning of parameters [32].

Given a protein sequence $X(i)(1 \leq i \leq n)$ of length $n$, specify an embedding dimension $m$ and a time delay $L$ to reconstruct the original sequence:

$$\begin{bmatrix} x(1) & x(1+L) & \cdots & x(1+(m-1)L) \\ \vdots & \vdots & & \vdots \\ x(j) & x(j+L) & \cdots & x(j+(m-1)L) \\ \vdots & \vdots & & \vdots \\ x(K) & x(K+L) & \cdots & x(K+(m-1)L) \end{bmatrix} j = 1, 2, 3, \ldots, K \tag{9}$$

Each row in the matrix can be regarded as a reconstructed subsequence, and there are $K$ reconstruction subsequence in total. The $j$-th reconstructed subsequence of $X(i)$ is $x(j), x(j+L), \ldots, x(j+(m-1)L)$. Sort in ascending order based on numerical value:

$$x(i + (j_1 - 1)L) \leq x(i + (j_2 - 1)L) \leq \ldots \leq x(i + (j_m - 1)L) \tag{10}$$

If the two values are equal, which is $x[i - (j_1 - 1)L] = x[i - (j_2 - 1)L]$, they are sorted according to the index $i$ of $j_i$. In this way, a subsequence $X(i)$ is mapped to $(j_1, j_2, \ldots, j_m)$. Therefore, each row in the matrix reconstructed by the protein sequence $X(i)$ can acquire a set of symbol sequences:

$$S(l) = (j_1, j_2, \ldots, j_m) \tag{11}$$

where $l = 1, 2, \ldots, k$, and $k \leq m!$, so every m-dimensional subsequence $X(i)$ is mapped to one of $m!$ permutations.

Through the above steps, we can represent the continuous *m*-dimensional subspace with a sequence of symbols, in which there are *m*!. The probability distribution of all symbols is represented by $P_1, P_2, \ldots, P_K$, where $K \leq m!$.

Finally, the permutation entropy of the protein sequence $w$ is calculated as:

$$H(m) = -\sum_{j=1}^{K} P_j ln P_j \tag{12}$$

For the convenience of calculation, we use 0 and 1 to represent the 20 amino acids of the protein sequence, as shown in Table 1. If the specified value of *m* is too small, the reconstructed sequence will contain too few states and the subsequences will lose validity and meaning. If the value of *m* is too large, the protein sequence will be homogenized, increasing the amount of calculation and failing to reflect the inherent subtleties of the protein sequence. Therefore, the embedding dimension *m* is generally 3 ~ 7, and $m = 5$ in this article. The influence of the delay time *t* can be ignored, usually $t = 1$. The overall calculation process is shown in Figure 1.

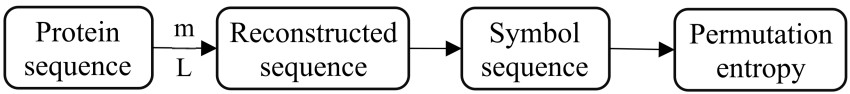

**Figure 1.** Calculation process of permutation entropy.

*2.4. Two Amino Acid Preferences*

On the basis of the above three entropies, we also added two amino acid propensity indicators to calculate the complexity of the protein sequence, namely Remark465 and Deleage/Roux given in the GlobPlot NAR article. We use Equation (11) to calculate the values of two amino acid preferences:

$$M_p(w) = \frac{1}{N} \sum_{j=1}^{N} w^p(j), p = 1, 2 \tag{13}$$

$w^p(j)$ represents the mapping value of the *p*-th amino acid preference, $p = 1, 2$ correspond to Remark465, Deleage/Roux, respectively, as shown in Table 2.

**Table 2.** Mapping values of amino acid sequences according to *p*-th preference.

|         | A       | R       | N       | D      | C       | Q       | E       | G       | H      | I       |
|---------|---------|---------|---------|--------|---------|---------|---------|---------|--------|---------|
| P = 1   | 0.1739  | −0.0537 | −0.2141 | 0.2911 | −0.5301 | 0.3088  | 0.5214  | 0.0149  | 0.1696 | −02907  |
| P = 2   | −0.2750 | −0.1790 | 0.4790  | 0.4625 | −0.1255 | −0.0550 | −0.2745 | 0.6675  | 0.1350 | −0.5150 |
|         | L       | K       | M       | F      | P       | S       | T       | W       | Y      | V       |
| P = 1   | −0.3379 | 0.1984  | −0.1113 | −0.8434| −0.0558 | 0.2627  | −0.1297 | −1.3710 | −0.8040| −0.2405 |
| P = 2   | −0.4385 | −0.0495 | −0.4765 | −0.4970| 1.1170  | 0.2965  | 0.1450  | −0.2570 | 0.0825 | −0.7055 |

*2.5. Preprocessing Process*

The prediction results after directly calculating all the above feature values for training are not ideal, so we use the sliding window to continuously intercept the area of the window length, calculate the five selected features, and assign them to all residues at the corresponding positions.

Given a protein sequence of length *N*, select a sliding window of length $L$ ($L < N$), and add $N/2$ zeros at both ends of the protein sequence. As the sliding window slides, calculate the mean value $V_i$ of the five-dimensional feature vector of each window, including Shannon entropy, topological entropy, permutation entropy, and two amino acid preferences, and assign $V_i$ to all residues in the window. Finally, Dividing the accumulated value of all

residues by the number of accumulations, the five-dimensional feature vector $X_i$ of each residue can be obtained:

$$X_i = \begin{cases} \frac{1}{j+L_0} \sum\limits_{i=1}^{j+L_0} V_i, 1 \leq j \leq L_0 \\ \frac{1}{N} \sum\limits_{i=j+L_0-L+1}^{j+L_0} V_i, L_0 < j \leq N - L_0 \\ \frac{1}{N_0-j-L_0+1} \sum\limits_{i=j+L_0-L+1}^{N_0-L+1} V_i, N - L_0 < j \leq N \end{cases} \quad (14)$$

Since the 995,189 residues in the DIS2209 data set are ordered, the 222,034 residues are disordered, and the number of positive and negative samples is unbalanced, we have added an oversampling method to increase the sample size and generate according to the law of samples with fewer categories. More samples of this label make the data tend to be balanced and the prediction results are more accurate. Compared with some existing oversampling schemes, we adopted SMOTE oversampling. The specific steps are as follows:

Step 1: For each sample of the minority class, use Euclidean distance as the standard to calculate the distance from all samples in the minority class sample set to obtain its k nearest neighbors.

Step 2: Set a sampling ratio according to the sample imbalance ratio to determine the sampling magnification $N$. For each minority sample $x$, randomly select several samples from its k nearest neighbors, assuming that the selected nearest neighbor is $x_{new}$.

Step 3: For each neighbor selected at random, it is defined as:

$$x_{new} = x + rand(0,1) \times |\tilde{x} - x| \quad (15)$$

## 3. Algorithm Scheme

For the problem of data imbalance, common processing methods include: sampling (over-sampling or under-sampling), cost-sensitive learning, and Ensemble learning methods. As mentioned above, we have adopted the method of oversampling to increase the number of samples in the minority class, so that the data tends to be balanced, but when the learner encounters this situation, it will encounter many repeated samples, so it will learn A special mode, which greatly increases the probability of overfitting. The ensemble learning algorithm is the result of merging multiple base classifiers, and fully considers the uncertainty and the possibility of misclassification of the sample.

Therefore, based on the ensemble learning (Boosting) algorithm, we used the oversampling method to preprocess the data and predict the data set DIS2209. Figure 2 shows the specific flow chart.

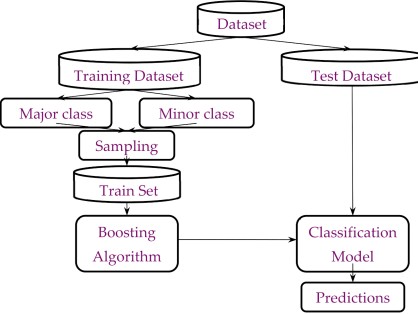

**Figure 2.** Scheme specific flow chart.

### 3.1. Gradient Boosting Decision Tree

Gradient Boosting Decision Tree (GBDT) is an algorithm to classify and regress data by using an additive model (that is, a linear combination of basis functions) and using the

negative gradient of the loss function to fit the approximate value of the current round of loss.

For a given sample set, first determine the cut point $c$:

$$f_0(x) = \arg\min_c \sum_{i=1}^{N} L(y_i, c) \tag{16}$$

For the number of iterations $m = 1, 2, \ldots, M$, assuming the number of samples $I = 1, 2, \ldots, N$, calculate the negative gradient:

$$r_{mi} = -\left[\frac{\partial L(y_i, f(x_i))}{\partial f(x_i)}\right]_{f(x)=f_{m-1}(x)} \tag{17}$$

Use $(x_i, r_{mi})$ to fit a regression tree to acquire the leaf node area $R_{mj}(j = 1, 2, \ldots, J)$ of the $m$th tree, where $J$ is the number of leaf nodes. For each sample in the leaf node, we find the smallest loss function, which is the best output value of the fitting leaf node:

$$c_{mj} = \arg\min_c \sum_{x_i \in R_{mj}} L(y_i, f_{m-1}(x_i) + c) \tag{18}$$

Strong learners updated to this round:

$$f_m(x) = f_{m-1}(x) + \sum_{j=1}^{J} c_{mj} I \left(x \in R_{mj}\right) \tag{19}$$

Finally acquire the learner expression:

$$f_M(x) = \sum_{m=1}^{M} \sum_{j=1}^{J} c_{mj} I \left(x \in R_{mj}\right) \tag{20}$$

By fitting the negative gradient of the loss function, we have found a general way to fit the loss error, so whether it is a classification problem or a regression problem, we can use GBDT to fit the negative gradient of the loss function Solve our classification regression problem. The only difference lies in the different negative gradients caused by different loss functions.

### 3.2. LightGBM

LightGBM is a framework that implements the GBDT algorithm. It is optimized on the traditional GBDT algorithm, which can speed up the training speed of the GBDT model without compromising the accuracy, and further improve the accuracy of predicting IDPS. The specific optimization is:

1. Using Histogram's decision tree algorithm, this algorithm can reduce memory usage and computing time through feature discretization.
2. Using the Leaf-wise algorithm with depth limitation, this strategy can split the same layer of leaves at the same time by traversing the data once, and it is easy to perform multi-thread optimization, and it is also easy to control the complexity of the model, and it is not easy to overfit.
3. The single-sided gradient sampling algorithm is used to exclude most of the samples with small gradients, and only the remaining samples are used to calculate the information gain. This algorithm can achieve a balance between reducing the amount of data and ensuring accuracy.
4. The use of mutually exclusive feature bundling algorithm can transform many mutually exclusive features into low-dimensional dense features, effectively avoiding unnecessary calculation of zero-value features.

5.     Supports efficient parallelism, including feature parallelism, data parallelism, and voting parallelism.

## 4. Performance Evaluation

We selected four indicators to evaluate the performance of the model: sensitivity (Sens), specificity (Spec), F1-Score (F1) and Matthews' correlation coefficient (MCC). Sens, Spec, and MCC are often used to evaluate prediction results in bioinformatics [33,34]. On this basis, we have added F1-Score to balance the accuracy and recall of the classification model. The following are the mathematical definitions of these four indicators:

Sensitivity:

$$Sens = \frac{TP}{TP + FN} \tag{21}$$

Specificity:

$$Spec = \frac{TN}{TN + TP} \tag{22}$$

F1-Score:

$$\frac{2 \times Precision \times Recall}{Precision + Recall} \tag{23}$$

where $Precision = TP/(TP + FP)$, $Recall = TP/(TP + FN)$.

Matthews correlation coefficient:

$$MCC = \frac{TP \cdot TN - FP \cdot FN}{\sqrt{(TP + FP) \cdot (TP + FN) \cdot (TN + FP) \cdot (TN + FN)}} \tag{24}$$

In all the above formulas, $TP$ represents the number of samples where the actual disordered residues are predicted to be disordered residues, $FP$ represents the number of samples where the actual ordered residues are predicted to be disordered residues, $TN$ represents the number of samples where the actual ordered residues are predicted to be ordered residues, $FN$ represents the number of samples where the actual disordered residues are predicted to be ordered residues.

## 5. Result and Discussion

### 5.1. The Effect of Permutation Entropy

In this article, we use permutation entropy to describe sequence complexity. Experiments have proved that low-complexity protein regions are often disordered. Permutation entropy has not been used to predict IDPS before, and it is used for the first time in our research. We calculated the permutation entropy of all ordered and disordered proteins in the data set DIS2209, and their probability density distribution is shown in Figure 3. It can be seen that there is a clear difference between ordered and disordered regions.

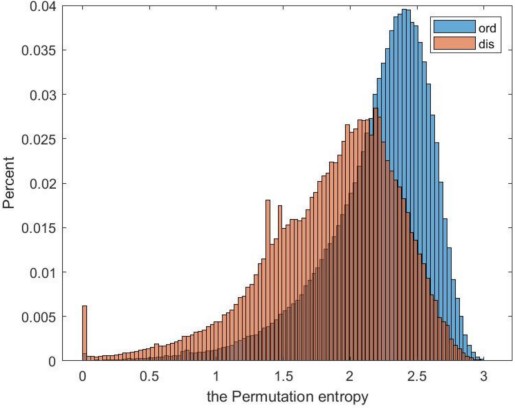

**Figure 3.** Probability distribution diagram.

In classification prediction, we can calculate the feature importance score of the predictive model, the score can highlight which features are important to the model and which features are not important to the model, helping us better understand the data and model. We show the feature importance of the five features in Figure 4. It can be seen that the feature importance of permutation entropy is higher than Shannon entropy and is basically the same as the preference of two amino acids.

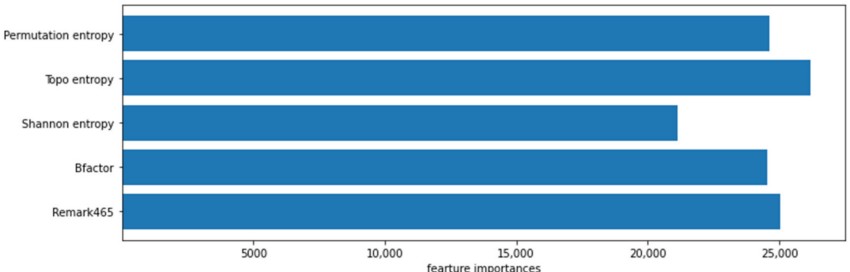

**Figure 4.** The importance of five features in the algorithm.

In order to ensure that permutation entropy plays a positive role in the prediction of intrinsically disordered proteins, we compared the prediction results with and without permutation entropy features in the two ensemble learning algorithms. The results are shown in Tables 3 and 4.

**Table 3.** The influence of permutation entropy on GBDT-PE.

|  | **Sens** | **Spec** | **F1** | **MCC** |
|---|---|---|---|---|
| Permutation entropy included | **0.766** | **0.789** | **0.565** | **0.463** |
| Permutation entropy not included | 0.744 | 0.760 | 0.528 | 0.413 |

**Table 4.** The influence of permutation entropy on LightGBM-PE.

|  | **Sens** | **Spec** | **F1** | **MCC** |
|---|---|---|---|---|
| Permutation entropy included | **0.781** | **0.862** | **0.651** | **0.526** |
| Permutation entropy not included | 0.725 | 0.816 | 0.569 | 0.465 |

By comparison, after adding permutation entropy to the two ensemble learning algorithms, the prediction results are significantly improved. In GBDT-PE, F1 and MCC have increased by 4% and 5%, respectively. In LightGBM-PE, the increase is the most obvious, F1 and MCC have increased by 9% and 6%, respectively. This is enough to show that permutation entropy plays a positive role in the prediction of inherent disordered proteins.

*5.2. The Influence of Sliding Window and Oversampling*

For the DIS2209 data set, we use a ten-fold cross-validation method to randomly divide the protein sequence into 10 subsets of roughly the same size, and use GBDT and LightGBM to train and predict data with different window sizes. The specific results are shown in Table 5.

By comparing various indicators, as the window length increases, the MCC value and F1 value gradually increase. When the window size is greater than 35, their values tend to be stable, so we choose a window length of 35 to process our features. The above-mentioned trend of change is shown in Figures 5 and 6. Similarly, we use GBDT and LightGBM to perform three oversampling schemes on the DIS2209 data set. When the SMOTE sampling scheme is used, the prediction effect is the best. The specific results are shown in Table 6.

**Table 5.** Performance comparison of different window sizes.

| Length | GBDT | | LightGBM | |
|---|---|---|---|---|
| | **F1** | **MCC** | **F1** | **MCC** |
| 11 | 0.4658 | 0.3842 | 0.5438 | 0.4281 |
| 15 | 0.4749 | 0.3958 | 0.5575 | 0.4323 |
| 19 | 0.4995 | 0.4132 | 0.5733 | 0.4431 |
| 23 | 0.5089 | 0.4246 | 0.5901 | 0.4561 |
| 27 | 0.5265 | 0.4448 | 0.6039 | 0.4671 |
| 31 | 0.5425 | 0.4584 | 0.6147 | 0.4786 |
| 35 | 0.5456 | **0.4624** | 0.6095 | **0.4868** |
| 39 | 0.5521 | 0.4610 | 0.6057 | 0.4767 |
| 43 | 0.5323 | 0.4489 | 0.5624 | 0.4648 |
| 47 | 0.5354 | 0.4421 | 0.5443 | 0.4608 |
| 51 | 0.5134 | 0.4312 | 05538 | 0.4615 |

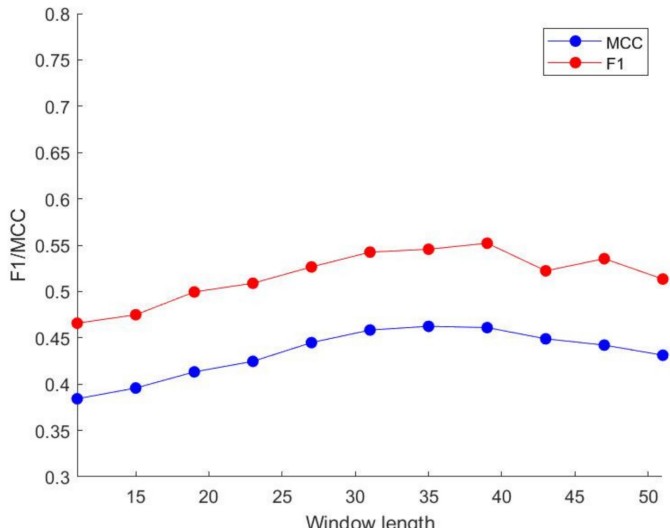

**Figure 5.** Different window size performance in GBDT.

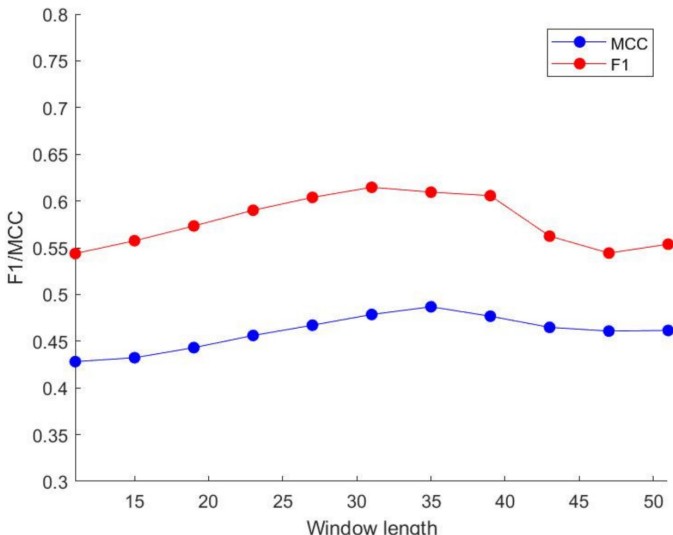

**Figure 6.** Different window size performance in LightGBM.

**Table 6.** Performance comparison of different oversampling schemes.

| Schemee | GBDT | | LightGBM | |
|---|---|---|---|---|
| | F1 | MCC | F1 | MCC |
| Not Sampled | 0.4343 | 0.4145 | 0.5622 | 0.4598 |
| Random Sampled | 0.5604 | 0.4554 | 0.6091 | 0.4729 |
| SMOTE Sampled | 0.5652 | **0.4652** | 0.6205 | **0.4816** |
| Synthetic Sampled | 0.5648 | 0.4632 | 0.6224 | 0.4799 |

Our pre-processing scheme improves the accuracy and stability of the prediction results of each learning method. Finally, we compare the MCC values before and after pre-processing in Table 7, taking sliding window size 35 and SMOTE oversampling as examples. We selected the Remark and Permutation Entropy features of all residues in the DIS2209 dataset and compared their performance before and after pretreatment at a window length of 31, as shown in Figure 7.

**Table 7.** The influence of the preprocessing scheme.

| | GBDT | LightGBM |
|---|---|---|
| Before pretreatment | 0.4143 | 0.4598 |
| After pretreatment | **0.4705** | **0.5258** |

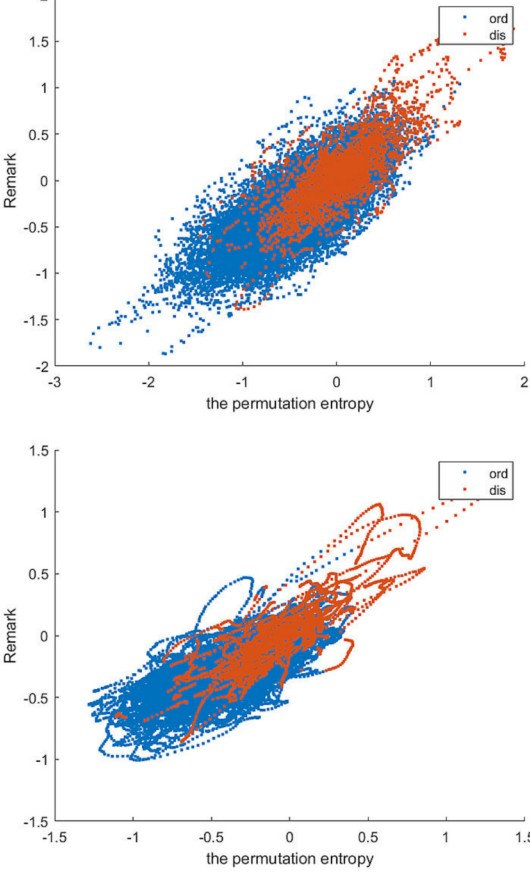

**Figure 7.** Remark465 and Permutation entropy before and after windowing.

In order to test the predictions of the model, we take the IAPP protein associated with type-2 diabetes from DisProt as an example and acquire the prediction result as Figure 8. At the same time, the standard prediction results are shown as Figure 9.

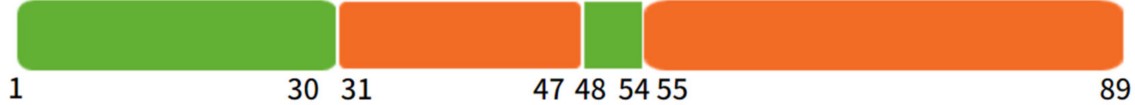

**Figure 8.** The prediction result of IAPP based on our system.

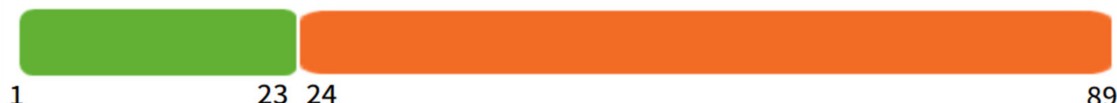

**Figure 9.** The prediction result of IAPP based on DisPort.

*5.3. Compare with Existing Forecasting Schemes*

In order to compare our scheme with the existing schemes, we used the dataset R80 collected by Yang et al. for testing. The R80 dataset contains 78 sequences with 29,243 ordered residues and 3566 disordered residues. Existing schemes include DISPRED2 [20], BVDEA [35], DisPSSMP [36], RONN [24], IsUnstruct [15], FoldIndex [14]. Table 8 shows the prediction results of each program.

**Table 8.** Prediction performance comparison based on test set R80.

|           | Sens      | Spec      | F1        | MCC       |
|-----------|-----------|-----------|-----------|-----------|
| GBDT-PE   | 0.774     | 0.791     | 0.566     | 0.471     |
| LightGBM-PE | 0.781   | 0.862     | **0.621** | **0.526** |
| DISOPRED2 | **0.972** | 0.405     | 0.482     | 0.470     |
| BVDEA     | 0.817     | 0.728     | 0.568     | 0.451     |
| DisPSSMP  | 0.767     | 0.848     | 0.605     | 0.463     |
| RONN      | 0.603     | **0.878** | 0.498     | 0.395     |
| IsUnstruct | 0.911    | 0.688     | 0.601     | 0.518     |
| FoldIndex | 0.488     | 0.811     | 0.342     | 0.224     |

Considering the classification method used, we use GBDT-PE and LightGBM-PE as the abbreviations of our scheme. Among these solutions, the highest Sens, Spec, F1 and MCC are DISPRED2, RONN, LightGBM-PE, LightGBM-PE, respectively. Only our LightGBM-PE scheme and IsUnstruct scheme have MCC values exceeding 0.5. Similarly, the F1 values of LightGBM-PE, DisPSSMP and IsUnstruct all exceed 0.6. After a comprehensive comparison, the results of our scheme are roughly the same as those of BVDEA, DisPSSMP and IsUnstruct. DisPSSMP and BVDEA need to calculate the 188 and 120 features of each residue in the protein sequence, respectively, while our solution only needs to calculate 5 features, which has lower computational complexity and simpler decision curve calculation, so our solution is more robust than DisPSSMP and BVDEA, and requires fewer learning samples.

## 6. Conclusions

In this paper, five features are selected to predict intrinsically disordered proteins, including Shannon entropy, topological entropy, permutation entropy and two amino acid preferences. Among them, permutation entropy is the first application in this field. In the data preprocessing stage, we used a sliding window to connect adjacent residues in the sequence. At the same time, we used the SMOTE oversampling scheme and two integrated learning algorithms to solve the imbalance of positive and negative samples in the original data. These schemes greatly improved Forecast accuracy. By comparing some existing schemes, our scheme has better F1 value and MCC value. The results show that the LightGBM-PE scheme can reach the highest MCC value of 0.526. Our solution uses only five features, has lower computational complexity, shorter training time, lower memory usage, and can adapt to training with a large number of data samples.

**Author Contributions:** Conceptualization, H.L.; project administration, H.L.; supervision, H.L.; validation, H.L. and X.Z.; formal analysis, X.Z.; investigation, X.Z.; methodology, X.Z.; software, X.Z.; writing—original draft preparation, X.Z.; writing—review and editing, X.Z.; resources, H.H. All authors have read and agreed to the published version of the manuscript.

**Funding:** The author(s) received no specific funding for this study.

**Data Availability Statement:** Publicly available datasets were analyzed in this study. These data can be found on the website: https://disprot.org (accessed on 6 October 2021).

**Conflicts of Interest:** The authors declare no conflict of interest.

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
