# Peer review of "Prediction of Intrinsically Disordered Proteins Using Machine Learning Based on Low Complexity Methods"

_algorithms, doi:10.3390/a15030086_

Round 1
Reviewer 1 Report
In this manuscript, He et. al. introduced their study of using a customized ML based method to predict IDPs. The manuscript is well written and the method and results are solid and promising. There are a few questions and concerns that remain to be answered by the authors.
- There are 5 features selected and engineered, including Shannon entropy, topological entropy, permutation entropy and two amino acid preferences. It would be nice if the authors could elaborate more the rationale behind picking each of these features specifically.
- Would removing any of the 5 features jeopardize the performance of the model?
- In addition to CV, would it be possible to have a separate independent external dataset for validation/testing purpose?
- In addition to LightGBM, would other ML methods improve the classification performance?
Reviewer 2 Report
This is an interesting manuscript. A few issues should be resolved before its publication.
In the introduction, the authors should mention a group of popular IDPs capable of amyloid-farming and are implicated in type 2 diabetes (10.1021/acs.chemrev.0c00981), Alzheimer's (10.1016/j.bbapap.2022.140767), Parkinson's (10.1038/s41531-021-00203-9), Prions (10.1021/acs.chemrev.1c00196).
Reading the article, the reader expects a test of the algorithm on IDPs such as a simple one like IAPP and a more complex one as alpha-synuclein and others. This gap should be filled to give a more applicative cut to the manuscript and consequently push researchers to use it.
Reviewer 3 Report
The paper presents and intrduces the permutaion (and other) techniques to identyfy the sequences attributed to intrinsically distorted proteins. The paper is addressed to „algorithms” people however the explanation of consecutive steps is not proper. Many symbols without explanation. Errors in equatios.
I find the main biological idea as interesting, however the form of presentation is unaceptable.
Table 1 – What is the purpose for this presentation ? I do not see any „hydrophobicity” information consumed in the whole text. Table 1 is useless.
Lines 114-117. The „u” is discussed but not explained U = ???? for the given example It is not given explicitly
Eq 4 How much it is (n-1)+1
Eq 5 = the symbol used unknown and not explained
Eq 7 – the index „l” not present in the expression
Eq 7 What it means (20n +n-1)+1
Eq 8 What is the intention to put „n+n-1+1-1”
What is the meaning of the 2 Mapping values not defined earlier
What it means „map hydrophobic residues to 0 or 1. Zero and 1 is the result of which calculation ? For which case ?
Table 2 nedds explanation – I do not see any significant progress.
The permutaion entropy – according to the text the only advantage to use it is to speed the calculation – What is the advantage of used method ?
Amino acid propensity scale – propensity to what ?
Figure 3 – If it is the main result – explanation as to the interpretation of the diagram shall be given
What is the purpose of this Figure – what it means „fearture importance” – term not defined
Figure 1 – to general – no relations to procedures discussed earlier in the paper
Table 4 – the results are not convincing – what was the example taken to get these results ?
The differences between permutation applied or not applied very small
The differences are not significant to call them better
Conclusions – The main advantage and sucess of this paper is to use the permutaion entropy for the first time. It is exressed few times in this paper. However I do not see any results How the DisProt data base got corrected after using all these protocols
In Summary
The paper in the version given is NOT ACCEPTABLE for publication
- The potential readers is a very narrow group of specialists – the biologists or any other life science specialists can not find any advantage of using this technique.
- To many symbols, equations without explanation
- Every symbol used in equation shall be explained – it is not the case
- The method is COMPLETE NOT REPRODUCTIVE – it is not possible to reconstruct the procedures described in the paper
- Generally speaking – TO MUCH CHAOS
Reviewer 4 Report
There are numerous editorial corrections that need to be made especially in the inconsistent capitalization of titles in the References on pages 13 and 14.
I am listing below several of these editorial corrections that need to be made:
Page 2: lines 76 and 77 and 85: commas needs to be inserted in numerical values so appear as: 1,217, 223 in line 76, 222,034 in line 77 and 1,217, 223 in line 85.
Page 5: line 176 heading should be "Preprocessing process".
Pages 13 and 14: The following references use only first word of title as capital with the other words in lower case: 1, 2,3,7, 9, 10, 12, 14, 15, 16, 17, 18, 19, 21, 22, 23, 31 and 32. This differs from the capitalization format for the following references in which capitalization is used for all words in title except for conjunctions such as "and", "for", etc.: 4, 5, 8, 11, 13, 20, 24, 25, 26, 27, 28, 29, and 30. Not only is this inconsistent capitalization for the Titles of the articles, but also there are instances of inconsistent capitalization of Journal titles, for example: References 14 and 29 have all lower case of words in Journal titles except for first word.
The above indicates that this manuscript needs a very careful editorial review before it can be accepted for publication in journal.
Round 2
Reviewer 2 Report
The authors' response is not satisfactory at all. Alpha-synuclein is not involved in Alzheimer's disease, the protein involved in this disease is Abeta. So the authors should first read the articles I suggested and then implemented their manuscript. It is unclear what the authors report with IAPP, perhaps the protein involved in type 2 diabetes? It is not clear in the revised version of the manuscript. The authors should correct the new part inserted in the introduction by reporting the appropriate references. Figures 8 and 9 are not adequately commented.
I hope that in the new version, the authors will consider my suggestions.
Round 3
Reviewer 2 Report
In the present form the manuscript is publishable.